Diversity, abundance, and host relationships of avian malaria and related haemosporidians in New Mexico pine forests

Marroquin-Flores Rosario A. 1
Williamson Jessie L. 1
Chavez Andrea N. 1 2
Bauernfeind Selina M. 1
Baumann Matthew J. 1
Gadek Chauncey R. 1
Johnson Andrew B. 1
McCullough Jenna M. 1
Witt Christopher C. cwitt@unm.edu 1
Barrow Lisa N. lnbarrow@unm.edu 1
1 Museum of Southwestern Biology and Department of Biology, University of New Mexico , Albuquerque , NM , United States of America
2 Bureau of Land Management Rio Puerco Field Office , Rio Puerco , NM , United States of America
Braga Erika
Electronic publication date: 2017 Aug 17
Publication date: 2017
Volume: 5
Electronic Location ID: e3700
Received 2017 May 29; Accepted 2017 Jul 26
Copyright: ©2017 Marroquin-Flores et al.
Copyright year: 2017
Copyright holder: Marroquin-Flores et al.
License: This is an open access article distributed under the terms of the Creative Commons Attribution License, which permits unrestricted use, distribution, reproduction and adaptation in any medium and for any purpose provided that it is properly attributed. For attribution, the original author(s), title, publication source (PeerJ) and either DOI or URL of the article must be cited.
License URL: https://creativecommons.org/licenses/by/4.0/

Keywords: Pinyon-juniper woodland, Birds, Apicomplexa, Host-parasite relationships, Ponderosa pine forest, Haemoproteus, Leucocytozoon, Plasmodium, Wildlife disease, Haemosporida

Funding: Bureau of Land Management Rio Puerco Field Office NSF DEB-1146491 PREP/FlyBase Fellowship NIH 5R25HG007630 NSF Postdoctoral Research Fellowship in Biology NSF PRFB-1611710 This work was supported by the Bureau of Land Management Rio Puerco Field Office (via the Colorado Plateau Cooperative Ecosystems Studies Unit agreement) and NSF DEB-1146491 to CCW. RAM-F was supported by a PREP/FlyBase Fellowship (NIH 5R25HG007630) and LNB was supported by the NSF Postdoctoral Research Fellowship in Biology (NSF PRFB-1611710). The funders had no role in study design, data collection and analysis, decision to publish, or preparation of the manuscript.

==============================
Avian malaria and related haemosporidian parasites (genera Haemoproteus, Plasmodium, and Leucocytozoon) affect bird demography, species range limits, and community structure, yet they remain unsurveyed in most bird communities and populations. We conducted a community-level survey of these vector-transmitted parasites in New Mexico, USA, to describe their diversity, abundance, and host associations. We focused on the breeding-bird community in the transition zone between piñon-juniper woodland and ponderosa pine forests (elevational range: 2,150–2,460 m). We screened 186 birds representing 49 species using both standard PCR and microscopy techniques to detect infections of all three avian haemosporidian genera. We detected infections in 68 out of 186 birds (36.6%), the highest proportion of which were infected with Haemoproteus (20.9%), followed by Leucocytozoon (13.4%), then Plasmodium (8.0%). We sequenced mtDNA for 77 infections representing 43 haplotypes (25 Haemoproteus, 12 Leucocytozoon, 6 Plasmodium). When compared to all previously known haplotypes in the MalAvi and GenBank databases, 63% (27) of the haplotypes we recovered were novel. We found evidence for host specificity at the avian clade and species level, but this specificity was variable among parasite genera, in that Haemoproteus and Leucocytozoon were each restricted to three avian groups (out of six), while Plasmodium occurred in all groups except non-passerines. We found striking variation in infection rate among host species, with nearly universal infection among vireos and no infection among nuthatches. Using rarefaction and extrapolation, we estimated the total avian haemosporidian diversity to be 70 haplotypes (95% CI [43–98]); thus, we may have already sampled ∼60% of the diversity of avian haemosporidians in New Mexico pine forests. It is possible that future studies will find higher diversity in microhabitats or host species that are under-sampled or unsampled in the present study. Fortunately, this study is fully extendable via voucher specimens, frozen tissues, blood smears, parasite images, and documentation provided in open-access databases (MalAvi, GenBank, and ARCTOS).

Introduction

Parasites are influential components of biotic communities, yet the vast majority of parasite diversity remains undescribed (Dobson et al., 2008; Poulin, 2014). A striking example is provided by the haemosporidian parasites (Protozoa: Apicomplexa: Haemosporida) that infect primates, rodents, bats, lizards, and birds. Avian malaria and related haemosporidians of the genera Haemoproteus (including Parahaemoproteus), Plasmodium, and Leucocytozoon are known to affect bird community structure (Atkinson et al., 2013; Kulma et al., 2013; Clark, Clegg & Lima, 2014), immune function (Atkinson et al., 2001; Beadell et al., 2007), telomere length and senescence (Asghar et al., 2015), survivorship (Atkinson et al., 2000), and fecundity (Knowles, Palinauskas & Sheldon, 2010). Over 200 avian haemosporidian species have been described based on morphology (Valkiūnas, 2005), but mitochondrial (mtDNA) sequences have revealed that at least one order of magnitude higher diversity exists (Bensch, Hellgren & Pérez-Tris, 2009; Clark, Clegg & Lima, 2014). Nearly two decades since the introduction of mtDNA ‘barcode’ survey methods (Bensch et al., 2000), many geographic regions and the vast majority of avian populations remain unsurveyed for haemosporidians. New community-level surveys will be critical to understanding their diversity, biogeography, and coevolutionary dynamics.

The need for new descriptive data on avian haemosporidian communities is vital, particularly in under-sampled regions and habitats, for several reasons. Interacting bird, dipteran, and avian haemosporidian species underlie the disease transmission cycle (Valkiūnas, 2005; LaPointe, Goff & Atkinson, 2010), and these species are likely to be susceptible to range shifts driven by climate warming. This situation creates the potential for novel host-parasite interactions. When naïve hosts encounter novel haemosporidian parasites, the consequences can be severe, as illustrated by the decimation of native Hawaiian honeycreepers after the introduction of Plasmodium relictum (Warner, 1968; Van Riper et al., 1986; Atkinson et al., 2000). Furthermore, increased temperatures can result in multiple reproductive cycles for the dipteran definitive hosts (Robinet & Roques, 2010), potentially causing increased infection risk or parasitemia, with negative consequences for bird population growth (Scott et al., 1983; Brown et al., 2001; Garamszegi, 2011). Increased contact between hosts and parasites may also facilitate host-switching, which appears to be a common mode of diversification in this group (Ricklefs & Fallon, 2002; Galen & Witt, 2014; Ricklefs et al., 2014). Descriptions of avian haemosporidian communities will elucidate the ecological niches, host relationships, and host-switching potential of parasite lineages, providing information that will be critical for wildlife management and will provide a basis for predicting climate change impacts.

The southwestern United States, in particular, is mostly unsurveyed and is likely to harbor a distinct avian haemosporidian assemblage, in part because its arid environment imposes challenges for the dipteran definitive hosts that serve as vectors (Yohannes et al., 2005; Lachish et al., 2011). The few previous community-level surveys of avian haemosporidian parasites in western North America have been conducted in California (Martinsen et al., 2008; Walther et al., 2016) and Alaska (Loiseau et al., 2012; Oakgrove et al., 2014). Here we report on the first community-level avian haemosporidian survey in New Mexico, USA. New Mexico’s arid climate and broad elevation gradients provide a compelling and untapped system in which to investigate avian haemosporidian diversity and ecology. We focus specifically on the breeding-season community in the elevational zone between 2,150–2,460 m, which is characterized by the transition from forests dominated by piñon pine to those dominated by ponderosa pine. Our objectives were: (1) To compare infection rates for each of the three avian haemosporidian genera (Haemoproteus, Plasmodium, and Leucocytozoon) among a suite of breeding bird species using microscopy and mtDNA; (2) To describe associations between avian haemosporidian haplotypes and their host species in a phylogenetic context; (3) To evaluate haplotype-richness (α-diversity) of the avian haemosporidian community in a previously unsurveyed region and habitat, including the proportion of lineages that are novel (never found in previous surveys). The survey results that we report are fully extendable via voucher specimens, frozen tissues, blood smears, parasite images, and documentation in open-access databases (MalAvi, GenBank, and ARCTOS).

Methods

Field sampling

We conducted fieldwork during June and July 2016 at three sites in northern New Mexico within the jurisdiction of the Rio Puerco Field Office of the Bureau of Land Management (BLM), an agency within the United States Department of Interior. The three sites included: (1) Mesa Chivato (McKinley and Sandoval Counties; on the northern flank of Mt. Taylor); (2) El Malpais National Conservation Area (Cibola County; on the southern side of the Zuni Mountains); and (3) Elk Springs (Sandoval County; on the western slope of the Jemez Mountains; Fig. 1). Sampling was conducted within a narrow elevational band (2,150–2,460 m) at the upper elevational extent of piñon-juniper woodland, where it transitions to ponderosa pine forest. These pine-dominated habitats were interspersed with patches of grassland and occasional Gambel oak, Douglas fir, or aspen. Permanent water was scarce in the sampled habitats, consisting of a tiny, spring-fed creek in the Elk Springs site, a natural spring (Ojo de los Indios) that has been developed in Mesa Chivato, and a few widely-dispersed watering troughs and earthen tanks for cattle or wildlife in Mesa Chivato and El Malpais National Conservation Area.

Figure 1 Map of study areas and 2016 sampling localities.

Fieldwork was conducted in three sites located in piñon-juniper and ponderosa pine woodland habitats (elevational range: 2,150–2,460 m). The number of infections detected by PCR and sequencing at each site for Haemoproteus (H), Plasmodium (P), and Leucocytozoon (L) is shown. Note that two infections from El Malpais could not be assigned to H or P because of poor sequence quality, resulting in 81 infections shown. Elevation is based on the SRTM Digital Elevation Database (Jarvis et al., 2008).

We focused on the breeding-season community in order to characterize locally-transmitted parasites. Sampling during the breeding season may also maximize detection because breeding birds frequently exhibit relapses of latent avian malaria infections (Applegate, 1970; Valkiūnas et al., 2004; Garvin & Schoech, 2006), possibly associated with increases in glucocorticoid stress hormones (Romero, 2002). Blood smears were prepared at the time of collection and were later fixed and stained in the lab (details below). Whole avian specimens were collected by mist-net or shotgun, preserved on dry ice, and transported to the Museum of Southwestern Biology (MSB) at the University of New Mexico for specimen preparation and preservation of tissues for genetic analysis. All samples were collected under Institutional Animal Care and Use Protocol 16-200406-MC and appropriate state and federal scientific collecting permits (New Mexico Department of Game and Fish Authorization Number 3217; US Fish and Wildlife Permit Number MB094297-0). Complete details on each specimen, including precise locality, collection method, and necropsy data are available in Table S1 and its embedded links to the ARCTOS database. Additionally, all novel haemosporidian haplotypes, host species infected, and occurrence sites were documented in GenBank, the MalAvi database (Bensch, Hellgren & Pérez-Tris, 2009) and in Tables S1–S2.

Genetic data collection

We extracted genomic DNA from frozen pectoral muscle tissue of 186 avian specimens using a QIAGEN DNeasy Blood and Tissue Kit, following the manufacturer’s protocol. To maximize detection of different parasite genera, we used three nested polymerase chain reaction (PCR) protocols to amplify a 478 base pair fragment of cytochrome b (cytb) in the haemosporidian mitochondrial genome, as described by Hellgren, Waldenström & Bensch (2004) and Waldenström et al. (2004). We used the outer primer pairs HaemNFI/HaemNR3 and HaemNF/HaemNR2 with the nested primer pair HaemF/HaemR2 to screen for Haemoproteus and Plasmodium. We used the outer primer pair HaemNFI/HaemNR3 with the nested primer pair HaemFL/HaemR2L to screen for Leucocytozoon. Each outer PCR contained 1.25 U AmpliTaq Gold DNA Polymerase (Applied Biosystems), 1× PCR Buffer II, 2.5 mM MgCl2, 0.2 mM dNTP, 0.5 µM each primer, and 20 ng template DNA in a total reaction volume of 25 µl. The thermal profile of this reaction was modified following Galen & Witt (2014) and consisted of an initial 8-min denaturation step at 95 °C followed by 20 cycles of 94 °C for 30 sec, 50 °C for 30 s, and 72 °C for 45 s, with a final 10-min extension at 72 °C. The nested PCR used the outer PCR product as the template (1 µl for Haemoproteus and Plasmodium; 2 µl for Leucocytozoon). Reaction conditions were the same for nested PCR except the number of cycles was increased to 35. Negative and positive controls were included in each PCR reaction to check for contamination and to verify successful DNA amplification. All PCR reactions were visualized on 2% agarose gels using SYBR Safe Gel Stain (Invitrogen, Carlsbad, CA, USA) to identify positive samples and verify the presence of PCR product of the expected length. All successful amplifications were purified using ExoSap-IT (Affymetrix, Inc., Santa Clara, CA, USA) and sequenced in both directions using dye terminator cycle sequencing on an ABI 3130 sequencer at the UNM Molecular Biology Core Facility.

Microscopic examination

Blood smears were air dried in the field and, within six months, were fixed using absolute methanol and stained for 50 min with phosphate-buffered Giemsa solution (7.0 pH). We examined each blood smear for evidence of haemosporidian blood parasites using either a Leica DM5000 B or a Nikon Labophot-2 light microscope, following identification protocol described by Valkiūnas (2005). We scanned at least 10,000 erythrocytes in all viable smears at 1,000×magnification using an oil immersion lens. We did not attempt to identify gametocytes to morphospecies; rather, we took digital photographs to archive in the ARCTOS database. We re-screened 76 (45%) of the blood smears to confirm negative or positive identifications after an initial comparison with PCR results.

Genetic data analysis

Parasite sequences were edited and aligned using the default alignment algorithm in Geneious version 8.0 (Biomatters Ltd; Kearse et al., 2012). We compared our sequences to previously sequenced infections in the public databases GenBank (National Center for Biotechnology Information, US National Library of Medicine) and MalAvi (Bensch, Hellgren & Pérez-Tris, 2009) using the Basic Local Alignment Search Tool (BLAST). We used the closest match to determine the parasite genus for each haplotype. Studies have indicated that avian haemosporidian sequences differing by a single base pair can differ in host association and in transmission (Bensch, Hellgren & Pérez-Tris, 2009). We therefore characterized parasite haplotypes differing by one or more base pairs from existing sequences in the GenBank and MalAvi databases as novel and named them following MalAvi naming conventions (first three letters of the genus and species of the first bird host species from which the haplotype was sequenced, followed by a haplotype number for that bird species). Some authors have suggested combining haplotypes into ‘lineages’ based on a 1% divergence rule (Outlaw & Ricklefs, 2014) and considering geographic distributions and hosts infected (Svensson-Coelho et al., 2013). By these definitions, ‘lineages’ are considered to represent putative species; however, species limits are difficult to determine with the data at hand. In this study, we tentatively treat each haplotype as a unique lineage; additional sampling will be required to determine whether some of these closely related haplotype groups may represent segregating variants within single populations.

In addition to reporting the proportion of infected birds, we reported the combined infection rate, defined as the number of infections detected divided by the number of birds screened. The latter metric accounts for the total number of infections in a host population, including multiple infections within a single host. We defined co-infection as testing PCR positive for more than one genus of haemosporidian parasite (i.e., possessing both Leucocytozoon and Haemoproteus/Plasmodium), or testing positive for more than one haplotype within a parasite genus (i.e., two distinct Haemoproteus or Plasmodium haplotypes), either in separate nested PCR reactions or by presence of double peaks in sequence chromatograms.

We estimated the phylogenetic relationships among New Mexico haemosporidian parasites based on cytb using maximum likelihood in RAxML version 8.2 (Stamatakis, 2014). Given the modest size of the dataset, we analyzed all codon positions as a single partition. We used the GTR+G model of nucleotide substitution and conducted a rapid bootstrap analysis with 1000 bootstrap replicates, after which we searched for the best-scoring maximum-likelihood tree. We did not specify an outgroup for the ML analysis and instead rooted the tree at the Leucocytozoon clade based on the relationships for Haemosporida determined using several loci and taxa (Borner et al., 2016). We generated a phylogenetic tree for the avian species sampled using BirdTree.org, which uses calibrated backbone trees of well-supported avian clades and generates trees for all bird species by partially constraining them to their respective clade (expanded methods in Jetz et al., 2012; Jetz et al., 2014). We used the phylogeny subsets tool to download a tree including only the species we sampled from the ‘Ericson All Species’ source of trees (Ericson et al., 2006).

Estimates of lineage diversity

We used EstimateS version 9.1.0 (Colwell, 2013) to generate an estimate of undiscovered lineage diversity present in northern New Mexico avian haemosporidian communities. This approach estimates species richness in a community based on rarefaction and extrapolation of reference samples (Colwell et al., 2012). We used counts for each parasite haplotype as individual-based abundance data regardless of host species identity, which should result in a conservative estimate of species richness. Rarefaction was conducted with 100 randomizations and the rarefaction curve was extrapolated with unconditional 95% confidence intervals to a total of 400 individuals, at which point the species richness curve reached an asymptote.

Results

Parasite abundance

We collected 186 individuals from 49 species and representing 19 families of New Mexico birds (Table S1). Twenty-six species tested positive for one or more of the three genera of avian haemosporidian parasites. In total, 65 out of 186 birds (34.9%) were infected based on PCR. These include 39 birds infected with one or more lineages of Haemoproteus (20.9%), 15 birds infected with Plasmodium (8.0%), and 25 birds infected with Leucocytozoon (13.4%). Two additional individuals tested positive for either Haemoproteus or Plasmodium in the PCR screening, but we were unable to identify these lineages to genus because of poor sequence quality. Combined infection rates were variable among parasite genera, as well as among host clades and host species (Fig. 2). A total of 18 (9.7%) individuals were co-infected, including two mixed infections comprised of Leucocytozoon with Plasmodium, 12 of Leucocytozoon with Haemoproteus, two of Haemoproteus with Haemoproteus, and one of Plasmodium with Plasmodium.

Figure 2 Phylogeny of haemosporidian haplotypes found in New Mexico birds.

Columns represent host clades (left) and host species (right). Dotted line indicates non-monophyly of non-passerines, and clade names are based on the names and topology from Moyle et al. (2016). Host species phylogeny was generated from http://BirdTree.org and the colors of host clade branches correspond to host species in each clade. Bar plots depict the combined infection rate (number of infections divided by number of birds screened) for each parasite genus: Haemoproteus (turquoise), Plasmodium (dark brown), Leucocytozoon (light brown). Stars indicate novel haemosporidian haplotypes. The parasite phylogeny was estimated in RAxML and branch labels indicate bootstrap values. The table is shaded to indicate which clade/species was infected with each haplotype. The number of infections sequenced for each haplotype and bird clade/species is shown, representing a total of 77 sequenced infections.

We completed microscopic examination for 168 individuals (90%) that had blood smears of adequate quality. We detected evidence of positive haemosporidian infection in 43 (25.6%) of the individuals screened. The rate of detection with PCR was higher than microscopy (Table 1). In 21 cases, PCR was positive with a negative microscopy result, and in three cases, microscopy was positive with a negative PCR result. Parasitemia (defined as the proportion of red blood cells infected out of 10,000) was <1% for the majority of slides examined. The highest level of infection was in an individual of Empidonax oberholseri (Tyrannidae) with ∼2% of red blood cells infected. Combining both PCR and microscopy results, 68 of 186 (36.6%) birds were infected.

Table 1 Positive infections identified from microscopy, PCR, and both methods combined.

Total screened indicates the number of individual birds screened by each method. Of those screened, the number and proportion of individuals that were positive are reported. Detection rates were higher for PCR, although three samples were positively identified using microscopy but not PCR.

		Haemoproteus/ Plasmodium	Leucocytozoon	Overall	
Method	Total screened	No. positive (%)	No. positive (%)	No. positive (%)	
Microscopy	168	40 (23.8%)	7 (4.2%)	43 (25.6%)	
PCR	186	55 (29.6%)	25 (13.4%)	65 (34.9%)	
Combined	186	58 (31.2%)	25 (13.4%)	68 (36.6%)	

Parasite diversity

We identified a total of 83 positive PCR infections, and obtained unambiguous sequences from 77 of them. Six sequences of poor quality were excluded from the parasite phylogeny and lineage diversity analyses because they could not be assigned to a haplotype. Four of these excluded sequences were positively identified as Leucocytozoon by the primer pair used, and two were either Haemoproteus or Plasmodium. These infections were included for the calculation of overall and Leucocytozoon infection rates. The 77 sequenced infections consisted of 43 distinct parasite haplotypes, including 25 Haemoproteus, six Plasmodium, and 12 Leucocytozoon haplotypes (Fig. 2; Table S2). Based on published sequences in the MalAvi and GenBank databases, 27 haplotypes (63%) identified were novel, which consisted of 17 novel haplotypes for Haemoproteus (meaning 68% of the haplotypes we found for the genus were novel), two for Plasmodium (33%), and eight for Leucocytozoon (67%). Additionally, we found evidence of infection in four juvenile birds; one with Leucocytozoon and two with novel Haemoproteus lineages, providing evidence for local transmission (Table S1). Sequences generated for each haplotype in this study are available on MalAvi and GenBank (Tables S1–S2; GenBank accession numbers: MF077648–MF077690).

Parasite phylogeny and host associations

The parasite phylogeny indicated strong support for the sister group relationship between avian Haemoproteus and Plasmodium (bootstrap value = 100). We recovered monophyletic relationships for each genus with strong to moderate support (bootstrap values: 100 for Leucocytozoon, 85 for Haemoproteus, 67 for Plasmodium). We found evidence for associations between host clades and parasite genera. All Haemoproteus haplotypes were restricted to three avian clades: Passerides clade 1b, Corvides, and Suboscines (Fig. 2). Similarly, all Leucocytozoon haplotypes were restricted to Passerides clade 1b, Passerides clade 2, and Corvides. Plasmodium infections occurred in all avian clades or groups sampled except non-passerines. We found no infections in non-passerine species, which may be due to low sample size (n = 8). Although node-support values were modest, some monophyletic Haemoproteus groups appear to be restricted to single avian clades (Fig. 2). Notably, the clade containing VIRPLU04 to VIGIL07 included 14 infections, all of which we recovered from avian hosts in the genus Vireo.

Estimates of lineage diversity

A rarefaction curve generated in EstimateS using the 77 infections and 43 haplotypes identified suggested that the total haplotype richness is ∼70 (95% CI [43–98]; Fig. 3). According to this method, we have identified ∼60% of the lineage diversity present at these sites, and sampling a total of ∼240 infections should be sufficient to capture >95% of the lineage diversity in this avian haemosporidian community. Based on our PCR-derived infection rate of 34.9%, this projection suggests we will need to screen ∼690 birds, or ∼500 additional samples to adequately characterize the avian haemosporidian community of New Mexico pine forests. This estimate should be regarded as a conservative minimum estimate of the sampling needed, as explained below.

Figure 3 Estimate of haemosporidian lineage diversity in northern New Mexico based on EstimateS rarefaction and extrapolation using 77 avian haemosporidian infections and 43 distinct haplotypes.

The point indicates the reference sample, solid line the rarefaction, and dotted line the extrapolation. The analysis suggests that sampling approximately 240 total infections would capture >95% of the haemosporidian lineage diversity in this community. The total haplotype richness is estimated to be 70 (95% CI [43–98]).

Discussion

Haemosporidian abundance in New Mexico pine forest breeding bird communities

We detected high levels of infection in the first community-wide survey of blood parasites in New Mexico breeding birds, with over one third (36.6%) of individuals infected with at least one of the three parasite genera. This level of infection is comparable to community surveys in other parts of the US including California (39.8% of 399 birds; Walther et al., 2016), Alaska (53% of 903 birds; Oakgrove et al., 2014), and Missouri (38.6% of 757 birds; Ricklefs et al., 2005). Community-level surveys in other parts of the world vary widely in avian haemosporidian infection rates, from 17.4% of 2661 birds in Brazil (excludes Leucocytozoon; Fecchio et al., 2017), to 79.1% of 532 birds in east Africa (Lutz et al., 2015). In our New Mexico study, Haemoproteus was the most abundant parasite genus (20.9% of birds infected), followed by Leucocytozoon (13.4%), then Plasmodium (8.0%). This generic composition was strikingly different from that found by some previous studies in western North America. For example, Walther et al. (2016) found much higher Plasmodium infection rates compared to the other two genera in a California songbird community, and Oakgrove et al. (2014) found Leucocytozoon to be the most abundant genus in an Alaska survey. The time of year in which samples were collected may have contributed to these patterns. For example, Walther et al. (2016) sampled from April to January, while we focused our sampling efforts on the breeding season (June and July). Haemoproteus may be easier to detect by PCR over short timeframes because relapses are generally longer in Haemoproteus infections compared to Plasmodium (Valkiūnas, 2005). Another factor to consider is the relative absence of standing water in ponderosa pine forest and piñon-juniper woodland habitats and how differences in vector ecology may contribute to these patterns. For instance, simuliid black flies that transmit Leucocytozoon parasites commonly lay eggs in running water (Adler, Currie & Wood, 2004). Interestingly, Elk Springs, the site with a spring-fed creek, had a higher Leucocytozoon infection rate (32%) compared to Mesa Chivato (13%) and El Malpais National Conservation Area (10%), but this pattern remains to be confirmed with additional sampling.

The variation in infection rate that we detected among host species suggests intriguing avenues for further investigation. We uncovered extremely high infection and co-infection rates in two Vireo species, Vireo gilvus and V. plumbeus. Of 13 individuals collected, 12 (92%) were positive for either Haemoproteus or Leucocytozoon, and eight (61.5%) were co-infected. Walther et al. (2016) also identified high infection rates in Vireo gilvus (n = 11) and identified V. gilvus as the only study species to be co-infected with more than three parasite lineages. The high rates of infection and co-infection indicate that Vireo species will be important to investigate as potential reservoirs for Haemoproteus and Leucocytozoon parasites (e.g., Moens et al., 2016). In contrast, the three species of nuthatches (Sitta pygmaea, S. canadensis, S. carolinensis) in our survey were completely uninfected (n = 12). It is possible that immune function or ecological characteristics minimize infection in these species. For example, nuthatches are cavity-nesters, a characteristic that is hypothesized to reduce exposure time to vectors and result in lower infection rates (Fecchio et al., 2011; Svensson-Coelho et al., 2013; Lutz et al., 2015; Medeiros et al., 2015).

The number of positive infections we detected with PCR differed somewhat from microscopy results, consistent with previous studies that have compared the two methods (Valkiūnas et al., 2008; Moens et al., 2016). Differences between PCR and microscopy detection are expected for at least three reasons. First, PCR can identify a positive infection with fewer than a single parasite per one million host cells (Hellgren, Waldenström & Bensch, 2004), infections that are unlikely to be detected using standard microscopic examinations of 10,000–100,000 cells (Atkinson et al., 2000). Second, it is conceivable that infections detected by PCR may be abortive infections, which would not develop into gametocytes in the blood stream (Valkiūnas et al., 2013). Third, detection by PCR appears to be sensitive to tissue type, with higher detection probability for heart, liver, or pectoral muscle tissue compared to blood (Svensson-Coelho et al., 2016). We sampled pectoral muscle tissue, for which Svensson-Coelho et al. (2016) found fewer false negatives compared to other tissue types, although in that study, no tissue type detected every infection that was detected by at least one of the four tissue types.

Novel parasite diversity and apparent host clade associations

Our survey revealed high diversity of avian haemosporidian parasites in northern New Mexico including several novel lineages. Of the 43 haplotypes we sampled, 16 have previously been identified and published in MalAvi or GenBank. Nine of these have only been documented in the US, six of which have only been identified within the western US, suggesting restricted geographical ranges within continental North America for at least some lineages. Haemoproteus was the most diverse lineage in our study with 25 haplotypes identified, 17 of which were novel. Likewise, several studies in other parts of the world including Asia, Europe, and sub-Saharan Africa have documented higher lineage diversity in Haemoproteus compared to Plasmodium (reviewed in Clark, Clegg & Lima, 2014). Other surveys have found either Plasmodium (California: Walther et al., 2016; South America: Svensson-Coelho et al., 2013; Fecchio et al., 2017) or Leucocytozoon (Alaska: Oakgrove et al., 2014; eastern Africa: Lutz et al., 2015) to have higher diversity compared to other genera. We found more novel lineages (27 total, or 63%) than similar community-level surveys in California (40% novel; Walther et al., 2016) and Alaska (49% novel; Oakgrove et al., 2014) despite having a much smaller sample size thus far. In this study, we sampled 186 birds compared to 399 birds (Walther et al., 2016) and 913 birds (Oakgrove et al., 2014), and all three surveys sampled a similar number of host species (46–49).

The apparent host breadth and geographic range of the lineages we sampled provides evidence for some generalist parasites, mostly within Plasmodium. For example, LAIRI01 was found in four different avian clades in our study, and has previously been reported in thePhilippines, Ecuador, and Mexico (Silva-Iturriza, Ketmaier & Tiedemann, 2012; Levin et al., 2013). We found one occurrence of WW3, which is distributed across Africa, Europe, and other parts of the US (Waldenström et al., 2002; Bensch & Akesson, 2003; Hellgren et al., 2007). One Haemoproteus lineage, SIAMEX01, also appears to be wide-ranging across the US and has been identified in several avian hosts (Ricklefs & Fallon, 2002; Levin et al., 2013). The majority of avian haemosporidian lineages found in our study, however, seem to be host specific at the species or clade level. One example includes the six Haemoproteus and two Leucocytozoon lineages that were specific to Vireo species in our study, although one of these haplotypes, TROAED12, was first described from a different host species, the house wren (Troglodytes aedon; Galen & Witt, 2014). Three of the vireo-specific lineages in our study (VIGIL02, VIGIL05, and VIGIL07) were also identified in Vireo species in California (Walther et al., 2016). Interestingly, a clade of haemosporidians specific to two eastern Vireo species was identified by Ricklefs et al. (2005), but those haplotypes cannot be directly compared at present because a different portion of cytb was sequenced. Among the major host clades, lineage diversity was highest for Passerides clade 1b and Corvides. Most Haemoproteus and Leucocytozoon lineages were sampled from a single avian host clade, but additional sampling is needed to confirm the patterns of host specificity in New Mexico pine forest breeding bird communities.

Our estimates of total avian haemosporidian lineage diversity indicate that a substantial number of haplotypes, ∼25, remain to be sampled in these communities; this is likely to be an underestimate. Although there was evidence of host specificity, we could not account for host-species identity in our rarefaction procedure in EstimateS because sample sizes for individual host species were small and biased towards common species. Considering the preponderance of unsampled and under-sampled bird species in the community, the avian haemosporidian diversity in New Mexico pine forests is likely much higher than the estimate presented here. For the same reasons, uneven sampling among the three sites may have exacerbated our underestimation of diversity, particularly if there is species-turnover among mountain ranges.

Conclusions

We uncovered a diverse community of avian haemosporidian parasites in New Mexico pine forests, with the majority of infections representing novel mtDNA haplotypes. We found evidence for host-specificity at the level of avian species and clades. There was also striking variation in infection rates among avian species and clades, exemplified by near universal infection of vireos and absence of infection in nuthatches. This study underscores the need for further sampling in southwestern North America in order to discover the diversity of ecologically important parasites that are interacting with birds. Follow-up studies should extend on the open-data provided here to test our extrapolation of the total avian haemosporidian haplotype diversity, and to determine the extent to which the avian haemosporidian community varies among neighboring ‘sky island’ mountain ranges or at elevations above or below the zone sampled here.

Supplemental Information

Supplemental Information 1 Supplemental Tables

Click here for additional data file.

Supplemental Information 2 Raw data - Parasite sequence alignment

Click here for additional data file.

We thank Michael J. Andersen, Celina Aguilar, Becky Bixby, Sara V. Brant, Mariel Campbell, Spencer Galen, and Xena Mapel for support; George Rosenberg of the Molecular Biology Core Facility at UNM for assistance with sequencing; the US Fish and Wildlife Service and New Mexico Department of Game and Fish for providing permits; the Bureau of Land Management for logistical support and site access; and Xi Huang and two anonymous reviewers for comments on the manuscript.

Additional Information and Declarations

Competing Interests

Author Contributions

Animal Ethics

Field Study Permissions

Data Availability

The authors declare there are no competing interests.

Rosario A. Marroquin-Flores analyzed the data, wrote the paper, prepared figures and/or tables, reviewed drafts of the paper.

Jessie L. Williamson analyzed the data, wrote the paper, reviewed drafts of the paper.

Andrea N. Chavez reviewed drafts of the paper, conceived and designed the study; Collected and prepared specimens and specimen data.

Selina M. Bauernfeind, Matthew J. Baumann, Chauncey R. Gadek, Andrew B. Johnson and Jenna M. McCullough reviewed drafts of the paper, collected and prepared specimens and specimen data.

Christopher C. Witt wrote the paper, reviewed drafts of the paper, conceived and designed the study; Collected and prepared specimens and specimen data.

Lisa N. Barrow analyzed the data, wrote the paper, prepared figures and/or tables, reviewed drafts of the paper, conceived and designed the study.

The following information was supplied relating to ethical approvals (i.e., approving body and any reference numbers):

The University of New Mexico Institutional Animal Care and Use Committee provided full approval for this research (Protocol 16-200406-MC).

The following information was supplied relating to field study approvals (i.e., approving body and any reference numbers):

Scientific collecting activities were approved by New Mexico Department of Game and Fish (authorization number 3217) and US Fish and Wildlife Service (permit number MB094297-0).

The following information was supplied regarding data availability:

The raw data is uploaded as a Supplementary File. It is also available via GenBank with accession numbers MF077648–MF077690.

https://www.ncbi.nlm.nih.gov/popset?DbFrom=nuccore&Cmd=Link&LinkName=nuccore_popset&IdsFromResult=1215209281.

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
