# Peer review of "Diversity, abundance, and host relationships of avian malaria and related haemosporidians in New Mexico pine forests"

_PeerJ, doi:10.7717/peerj.3700_

## Round 0.1 · original submission · Minor Revisions

The reviewers and I myself consider that the manuscript contributes to the specific field since it describes considerable diversity of avian hemosporidians and highlights the occurrence of new lineages in New Mexico. In particular, I appreciated the inclusion of microscopic observation to evaluate the prevalence of avian hemosporidians. However, the manuscript requires some modifications before been accepted. Thus, please clarify all points raised by the three expert reviewers as detailed below. I strongly advice authors to include a supplemental material including the MalAvi names and accession numbers of the lineages recovered to make the data available to any reader. It would be interesting to make the discussion more robust and complete including the comments and suggestions pointed by the reviewers.

Reviewer 1 ·

Basic reporting

No comment

Experimental design

No comment

Validity of the findings

No comment

Additional comments

This manuscript reports a high diversity of avian haemosporidians in New Mexico, with a high proportion of new parasite haplotypes. Although they did not test any ecological hypothesis, these results provide insights on host-parasite interactions and open precedents for further studies at high altitudes in southeastern North America and also benefit the field of knowledge as a whole.

Well conducted study, with appropriate prevalence and diversity estimation as authors applied robust morphological and molecular methods. However, I do think that authors can address some points that I highlighted below:

Line 2: The term “Avian malaria parasites” have been attributed exclusively to the genus Plasmodium. For all three genera together the term “haemosporidian parasites” or “avian haemosporidians” has been widely used and prefered (see Valkiūnas & Iezhova, Malaria Journal, Mar 3;16(1), 2017 - Exo-erythrocytic development of avian malaria and related haemosporidian parasites.) I strongly suggest to consider this nomenclature throughout the manuscript.
Lines 14-17: Try to concatenate both sentences to make this idea shorter.
Line 54: Authors introduced the term “lineages” without providing any background on that. The present paper may reach a readership that is not familiar with this term, so would be important to define it in a previous sentence. Perhaps together with lines 36-39.
Line 94: Authors did not mention the method for bird sampling and I assume it was made by shooting flying birds. It is necessary to specify it in the Methods section.
Lines 106-107: Make it clear that you mean haemosporidian haplotypes.
Lines 116-118: Did authors find any difference in positive rates and parasite sequences obtained when using both outer primer sets? To make it clear, using different primer sets yielded different results for the same sample? This would be of great interest for future works in your sampling area and in other parts of the American continent.
Line 147: Please provide GenBank accession numbers for newly described lineages.
Lines 154 and 156: change “bp” to “base pair”.
Lines 197-202: These definitions should be in the Methods section.
Line 221: “(“ is italicized.
Lines 222-223: “We identified…”. This sentence should appear before the sentence starting line 217. Adding up the number of parasites in Fig. 1 gave 78 infections, not 77 as mentioned.
L 241: According to Fig. 2, all 14 infections from this mentioned clade were detected in Vireo birds. Is there something incorrect or this is actually a host-specific parasite clade?
Line 270-275: It is not clear what authors wanted to show here and I think it can be more elaborated on how these habitat differences would influence vector distributions for each group (Simuliidae, Ceratopogonidae and Culicidae). Furthermore, I cannot see the differences in rates of infection according to Fig. 1: rates of Haemoproteus were quite similar between Elk Springs and Mesa Chivato (50% and 63%, respectively, and definitely not higher for Elk Springs) and Leucocytozoon infection rates also were quite similar between all sites (43%, 30% and 29 for Elk Springs, El Malpais and Mesa Chivato, respectively). These differences are not intuitive, and to compare then authors may need to conduct an statistical analysis.
Lines 296-297: This third assumption is related to the first one, so I think it is not essential to the manuscript. With that, the sentence that follows (lines 297-298) can be removed as it does not provide information relevant to the context if authors decide to exclude the third assumption. However, lines 299-303 could constitute a third assumption.
Line 303: “perfect detection” is not informative.
Lines 308-310: Authors should make it clear that “five haplotypes” are within the group of “eight haplotypes” found exclusively in the US.
Lines 311-315: Clark et al (International Journal for Parasitology, 2014) mention a large number of surveys in which Haemoproteus is the most common haemosporidian. Would be better to acknowledge that if authors keep this part of the discussion.
Lines 315-318: Authors are comparing results, but there is no discussion regarding this topic although it is quite interesting. Moreover, the content in the last parenthesis is quite confusing.
Lines 336-337: I don’t understand what authors mean by “we were not able to account for host-species identity in our rarefaction procedure in EstimateS”. It seems that something is missing here.
Line 350: I think that “symbionts” is not appropriate as the type of interaction studied is parasitism.
Lines 564-565: I think that here should be something like “number of detected parasites”, as authors found less than 78 infected birds, with some being infected by more than one parasite.

Table 1: Including a Grand Total row below the PCR results would make easier to interpret your results.

Supplementary material: I strongly recommend authors to include parasite data (haplotype and MalAvi names and accession number) also in Table S1. It would be easier to see host-species and location for parasites as well, what can help future studies intending to compile datasets.

·

Basic reporting

This whole manuscript is written clearly and easy to read.
However the writing is a bit too wordy somewhere, especially the method part. Will be better if cut off some redundant descriptions.
The texts of host species in Fig. 2 is not very easy to read. Maybe try to rotate 180?

Experimental design

Research question and method description is clear.
Some places can be more briefly such as lines 112-119, can be rewritten as for example “To maximize detection of parasites from different genera, we used three nested polymerase chain reaction (PCR) protocols to amplify a 478 base pair (bp) fragment of cytochrome b (cytb) gene in the haemosporidian mitochondrial genome, as described by Hellgren, Waldenström & Bensch (2004) and Waldenström et al. (2004). We used the outer primer pair HaemNFI/HaemNR3 with the nested primer pair HaemF/HaemR2 to screen for Haemoproteus / Plasmodium and HaemFL/HaemR2L for Leucocytozoon, respectively.”

Validity of the findings

Questions were answered and the discussion fits the result.
Just a few comments:
Lines 222-225, which ones were excluded? Excluded in all the following analyses or only in constructing phylogenetic tree? Were they included in the 43 haplotypes?
Lines 267-270, as parasite infection varies across the year, can this difference be caused by different sampling season (e.g. Walther et al. (2016) sampled in April, May, June, October and January while your samples were taken in June and July)?
Lines 308-309, any information about the infected host species (i.e. may only occur in locally resident birds)?

Additional comments

This manuscript has expanded our knowledge on parasite diversity in a new area by doing a survey on community level. Many novel lineages as well as host-parasite assemblages were detected. It is quite interesting to estimate lineage diversity and find there are more to explore.
The main weakness is the slightly wordy writing that can be improved before acceptance.

Reviewer 3 ·

Basic reporting

The study reports on a small survey of Haemosporida in the high elevation forests of New Mexico. This is an important survey since these the relationships between Haemosporida and birds, an important model system, are completely uncharacterized.

The study is clearly written, nicely organized, and figures are suitable for publication. It could benefit from a more thorough review of existing literature (see specific comments), but in general, the literature review and background is adequate. The manuscript is self-contained.

Experimental design

This is a natural history survey of a site that is under-characterized. This aim and its importance are well defined in the manuscript. The approach to characterize the relationships between birds and Haemosporida follow established techniques in the literature.

Validity of the findings

The results fill a knowledge gap concerning the relationships between Haemosporida and a unique and under-sampled avifauna. The survey provides a solid foundation for future surveys and natural experiments on the relationships between parasites and hosts.

Additional comments

Line 285: “In contrast, the three species of nuthatches (Sitta pygmaea, S. canadensis, S. carolinensis) in our survey were completely uninfected (n = 12). It is possible that immune function or ecological characteristics such as cavity-nesting (Fecchio et al., 2011; Svensson-Coelho et al., 2013; Lutz et al., 2015) minimize infection in these species.”

It would appear that this could be improved by a more thorough discussion of the factors that might reduce prevalence in a bird species. For instance, cavity nesting is hypothesized to lower contacts with vectors. Other studies have shown more directly that reduced vector contacts are associated with lower prevalence of vector-borne disease. Please see:

Mendes, L., Piersma, T., Lecoq, M., Spaans, B., & E Ricklefs, R. (2005). Disease‐limited distributions? Contrasts in the prevalence of avian malaria in shorebird species using marine and freshwater habitats. Oikos, 109(2), 396-404.

Medeiros, M. C., Ricklefs, R. E., Brawn, J. D., & Hamer, G. L. (2015). Plasmodium prevalence across avian host species is positively associated with exposure to mosquito vectors. Parasitology, 142(13), 1612-1620.

MARTÍNEZ–DE LA PUENTE, J., Martínez, J., RIVERO-DE-AGUILAR, J., Del Cerro, S., & Merino, S. (2013). Vector abundance determines Trypanosoma prevalence in nestling blue tits. Parasitology, 140(08), 1009-1015.


Vireo infections

Vireos tend to have a specialized clade of Parahaemoproteus (see Ricklefs et al. 2005). This seems relevant to some of the discussion about clade specific Haemosporida infection and high rates of parasitism in the two vireo species. The manuscript would improve with a more explicit discussion of this information.

---

## Round 0.2 · accepted · Accept

All points raised by the reviewers and myself were properly addressed.

Reviewer 1 ·

Basic reporting

Authors addressed all points accordingly and the current version of the manuscript is suitable for publication.

Experimental design

No comment

Validity of the findings

No comment

·

Basic reporting

no comment

Experimental design

no comment

Validity of the findings

no comment

Additional comments

This MS is well designed and can expand our knowledge on avian haemosporidians in novel regions, the idea to estimate local diversity of parasite lineages is really nice. I believe that it is ready to be published.